# Overview of the Impact of Pathogenic LRRK2 Mutations in Parkinson’s Disease

**DOI:** 10.3390/biom13050845

**Published:** 2023-05-16

**Authors:** Genta Ito, Naoko Utsunomiya-Tate

**Affiliations:** Department of Biomolecular Chemistry, Faculty of Pharma-Sciences, Teikyo University, Tokyo 173-8605, Japan

**Keywords:** Parkinson’s disease, leucine-rich repeat kinase 2, LRRK2, mutation, pathology

## Abstract

Leucine-rich repeat kinase 2 (LRRK2) is a large protein kinase that physiologically phosphorylates and regulates the function of several Rab proteins. LRRK2 is genetically implicated in the pathogenesis of both familial and sporadic Parkinson’s disease (PD), although the underlying mechanism is not well understood. Several pathogenic mutations in the *LRRK2* gene have been identified, and in most cases the clinical symptoms that PD patients with LRRK2 mutations develop are indistinguishable from those of typical PD. However, it has been shown that the pathological manifestations in the brains of PD patients with LRRK2 mutations are remarkably variable when compared to sporadic PD, ranging from typical PD pathology with Lewy bodies to nigral degeneration with deposition of other amyloidogenic proteins. The pathogenic mutations in LRRK2 are also known to affect the functions and structure of LRRK2, the differences in which may be partly attributable to the variations observed in patient pathology. In this review, in order to help researchers unfamiliar with the field to understand the mechanism of pathogenesis of LRRK2-associated PD, we summarize the clinical and pathological manifestations caused by pathogenic mutations in LRRK2, their impact on the molecular function and structure of LRRK2, and their historical background.

## 1. Introduction

Parkinson’s disease (PD) is the second most common neurodegenerative disorder and is pathologically characterized by the selective loss of pigmented neurons in the substantia nigra pars compacta (SNpc) and in the locus coeruleus (LC) in the brainstem [1]. The presence of Lewy bodies (LBs) in the remaining neurons is another pathological hallmark of PD. Clinical manifestations of PD include resting tremor, rigidity, bradykinesia, and postural instability. Although two hundred years have passed since the first clinical description of PD by James Parkinson in 1817 [2], the mechanism underlying neurodegeneration and the pathological significance of LB formation remain largely unknown, and there is no disease-modifying therapy for PD.

The majority of PD cases are sporadic, but there are familial cases of inherited PD (familial PD; FPD); genetic analysis of the families has identified more than twenty loci as responsible for the pathogenesis of PD [3]. The genes mutated in FPD include *SNCA* (PARK1 (OMIM #168601) and PARK4 (OMIM #605543)) and *LRRK2* (PARK8 (OMIM #607060)). The *SNCA* gene encodes α-synuclein. Since missense mutations in the *SNCA* gene and its amplification cause PD [4,5,6,7,8], and since abnormally hyperphosphorylated α-synuclein is the major component of LBs [9,10,11], α-synuclein is thought to play an important role in the pathogenesis of both familial and sporadic PD. The *LRRK2* gene encodes Leucine-rich repeat kinase 2 (LRRK2), a large multidomain protein kinase in which several missense mutations have been linked with PD (Figure 1A) [12,13]. Recent work has shown that LRRK2 is physiologically involved in the phosphorylation of Rab proteins, small GTPases that regulate vesicular trafficking [14,15,16]. However, the pathological role of Rab phosphorylation by LRRK2 remains unknown. Importantly, both *SNCA* and *LRRK2* have been associated with an increased risk of developing sporadic PD in a number of genome-wide association studies [17,18,19,20], suggesting that α-synuclein and LRRK2 play an important role in the pathogenesis of PD.

In this review, we will summarize the effects of pathogenic mutations in LRRK2 on the clinical symptoms and the neuropathology of patients, in addition to their effects at the molecular and cellular levels, to provide a comprehensive overview of the pathological roles of LRRK2 in PD.

## 2. Amino Acid Substitutions in LRRK2 Genetically Implicated in the Pathogenesis of PD

Although several genetic studies have identified a large number of missense mutations in LRRK2 [22,23], the pathogenicity of these mutations remains largely unproven. Among these mutations, eight mutations (I1371V, N1437H, R1441C/G/H, Y1699C, G2019S, and I2020T) have segregated with PD in several FPD families and are thought to be pathogenic. In addition, the N1437D and R1441S mutations have each been found in one FPD family. Since these two mutations occur in the same residues as the pathogenic mutations mentioned above (N1437H and R1441C/G/H), it is highly likely that these two mutations are also pathogenic. The mutated residues are all highly conserved across species (Figure 1B), suggesting that these residues play an important role in maintaining the function and/or structure of the LRRK2 protein.

## 3. Clinical Manifestations and Brain Pathology of Patients with LRRK2 Mutations

This section summarizes the clinical manifestations and brain pathology, when available, of patients with LRRK2 mutations. An at-a-glance table summarizing this section is provided in Table 1.

### 3.1. I1371V Mutation

The I1371V mutation of the *LRRK2* gene was found by Paisán-Ruíz and colleagues in 2005 in an East Indian family with dominant inheritance of PD (Family PD4) [24]. The proband developed PD at the age of 41 years. Except for the relatively early onset, the clinical course of this patient is reported to be consistent with typical PD. The I1371V mutation was then found in two patients from an Italian family (Family MI-007) [25]. These patients developed the disease at the ages of 33 and 61 and responded well to levodopa. One of them developed severe cognitive impairment. The pathogenicity of this mutation was inconclusive in this report because the control group included a 55-year-old with the I1371V mutation who did not develop PD. Two other French families with the I1371V mutation were reported in 2009 (Family C and D) [26]. In total, there were three affected individuals with ages at onset of 48, 48, and 71 years, respectively. Their reported clinical course resembles that of typical PD, manifesting as levodopa-responsive parkinsonism without signs of dementia. Taken together, although the number of patients with the mutation is small, these familial cases suggest that the I1371V mutation causes typical but relatively young onset PD.

The neuropathology of a member of Family MI-007 described above has been reported [27]. The patient developed PD at the age of 61 with mild cognitive impairment in later years and died of a pulmonary embolism at the age of 71. In the brain, there was severe and moderate neuronal loss in the substantia nigra (SN) and the locus coeruleus, respectively. They also noted moderate neuronal loss in the dorsal glossopharyngeus-vagus complex. α-Synuclein-positive Lewy pathology was seen in the affected areas as well as in the cerebral cortex. Tau-positive neurofibrillary tangles were present only in the hippocampus and amygdala. The observed neuropathology was consistent with that of typical PD.

### 3.2. N1437H Mutation

The N1437H mutation was first described by Aasly and colleagues in 2010 [28]. They reported a large Norwegian family (F04) inheriting the heterozygous N1437H mutation in the *LRRK2* gene for four generations and an affected carrier with familial parkinsonism (F45) sharing the same haplotype. In these families, the segregation pattern was consistent with autosomal dominant inheritance [28]. The mean age at onset was 47 years (37–61 years), significantly younger than that of sporadic PD cases in Norway (58 ± 11 years). The clinical manifestations of affected individuals carrying the mutation have been reported to be largely indistinguishable from sporadic PD. Recently, a large Polish family inheriting the N1437H mutation has been reported [29]. They identified three affected family members carrying the mutation whose mean age at onset was 52 years (49, 52, and 55 years). Their clinical course was similar to that of sporadic PD: asymmetric onset of upper limb rigidity with good response to levodopa. A Swedish patient with the N1437H mutation denying familial PD was also reported [30], suggesting an incomplete penetrance of this mutation. The age at onset was 50 years, and the clinical course of this patient was again similar to that of typical PD. The patient did not show any cognitive deficits. In conclusion, the N1437H mutation is pathogenic but relatively rare, occurring mainly in Northern Europe and causing clinically typical PD with an earlier age of onset.

The neuropathology of the Swedish patient has been reported [30]. They observed an almost complete loss of melanin-containing neurons in the SNpc with a few LBs in the remaining pigment-containing neurons. Neuronal loss and α-synuclein pathology were observed in the LC as well as in the dorsal motor nucleus of the vagus. This result suggests that the N1437H mutation causes severe loss of dopaminergic neurons and Lewy pathology in the midbrain as observed in typical PD.

### 3.3. N1437D Mutation

The N1437D mutation was found in two Chinese FPD families (Family AD-242 and AD-023) in 2020 [31]. There were a total of three affected individuals carrying the mutation in these families. The mean age at onset of these patients was 47.5 years (44 and 51 years; not reported for one patient), a relatively younger onset consistent with the N1437H mutation. The clinical symptoms of these patients were not clearly described in this report. Neuropathology was also not reported.

### 3.4. N1437S Mutation

The N1437S mutation segregating with PD in a small family was identified by Haebig and colleagues in 2010 [32]. This is the only study reporting the N1437S mutation. There were apparently three affected individuals harboring the mutation, although the details, including their clinical manifestations and neuropathology, were not described.

### 3.5. R1441C Mutation

The family inheriting the heterozygous R1441C mutation (Western Nebraska family; Family D) was first reported by Wszolek and colleagues in 1995 [33] with a follow-up study in 2004 [34] before the responsible mutation in the *LRRK2* gene was identified. Family D is a large family from the USA with more than 20 affected individuals and a segregation pattern consistent with autosomal dominant inheritance. The mean age at onset was 65 years [34]. Their clinical manifestations, such as levodopa-responsive parkinsonism, were similar to those of typical PD, and dementia was not seen.

The results of the neuropathological examinations of the four family members were reported in the follow-up study and in the paper that first identified the R1441C mutation [13]: Loss of pigmented neurons in the SN and LC with gliosis was seen in all individuals. LBs were found in the remaining neurons in the SN and LC in two of the four individuals. In one individual without Lewy pathology, Tau deposition was observed in neurons and glial cells, while no LB or Tau deposition was observed in the other individual. Collectively, the R1441C mutation causes clinically typical late-onset PD but neuropathologically diverse neurodegeneration in the brainstem.

Following the identification of the R1441C mutation in the *LRRK2* gene, a number of families have been identified in the USA [35,36,37], Brazil [38], Italy [25], Belgium [39], and China [31]. The overall clinical manifestations of the R1441C mutation carriers were similar to those of Family D. Of note, haplotype analysis has shown that the Italian family [25] and one of the American families (Family 469 in [13]) share the same haplotype, suggesting that these two families have a common founder, whereas Family D and the Belgian families have different haplotypes [39]. This result suggests a multiple or very ancient origin of the R1441C mutation.

The R1441C mutation has also been identified in a number of case-control studies [38,40,41,42,43,44,45,46,47,48,49]. In their case-control study, Haugarvoll and colleagues reported that the clinical presentation of patients with the R1441C mutation was similar to sporadic PD, with a mean age at onset of 60 years [44]. They also showed that the R1441C mutation originated from multiple ancestors from different parts of the world, suggesting that the Arg1441 residue is a hotspot for mutation.

### 3.6. R1441G Mutation

The first report of families inheriting the heterozygous R1441G mutation was by Paisán-Ruíz and colleagues in 2004 in their paper reporting the discovery of PD-linked mutations in the *LRRK2* gene [12]. There were four families in the Basque region of Spain, three of which shared the same haplotype, indicating the ancestral relationship of these families. They further screened the mutation in 137 Basque PD patients, both sporadic and familial, and found 11 Spanish cases, 10 Basques, carrying the R1441G mutation, suggesting that the R1441G mutation is a common cause of PD in the Basque population. The clinical phenotype of PD patients carrying the R1441G mutation was briefly mentioned in the paper: mean age at onset around 65 years, levodopa-responsive parkinsonism, and absence of cognitive impairment. Another family carrying the R1441G mutation was reported shortly after the first paper [35]. They sequenced all exons of *LRRK2* in their FPD cases and identified the R1441G mutation in a Spanish family with seven affected individuals sharing the same haplotype as the original Basque families. The age at onset of the proband was 61 years. A Japanese family inheriting the R1441G mutation with five affected individuals has also been reported [50]. The ages at disease onset of the three affected individuals in this family were 28, 34, and 55 years, younger than typical PD and other R1441G familial cases. They noted intrafamilial clinical heterogeneity in this family, including variable severity of parkinsonism and occurrence of psychosis. The haplotype of this Japanese family was different from the original Basque families, supporting the notion that the Arg1441 residue is a hotspot for mutation.

Although the FPD cases inheriting the R1441G mutation are relatively rare, there have been many reports showing that the R1441G mutation is indeed a common cause of PD in Spain, mostly in the Basque region [38,42,51,52,53,54,55,56,57,58,59,60,61]. The motor symptoms of PD patients carrying the R1441G mutation are very similar to those of idiopathic PD (reviewed in [62]). Tijero and colleagues have shown that PD patients carrying LRRK2 mutations (G2019S or R1441G) have less autonomic dysfunction, such as orthostatic hypotension and less severe sympathetic denervation, as shown by meta-iodobenzylguanidine (MIBG) scintigraphy, when compared to idiopathic PD [63]. It has also been reported that R1441G- and G2019S-associated PD patients were less likely to have cognitive and neuropsychiatric impairment when compared to sporadic PD [64]. Hyposmia was also shown to be less common in R1441G-associated PD patients [65], suggesting that the symptoms associated with the R1441G mutation are relatively limited to motor aspects.

Ruiz-Martínez and colleagues have shown that the age-dependent penetrance of the R1441G mutation is estimated to be 83.4% at 80 years of age [59]. The age of onset varies among siblings, suggesting that additional genetic or environmental factors should contribute to the onset of PD symptoms by the R1441G mutation (reviewed in [66]).

Several neuropathological studies have been performed in patients carrying the R1441G mutation. The first report was of a Spanish patient carrying the mutation who developed symptoms at the age of 68 and died at the age of 86 [67]. Neuropathological examination showed approximately 60% loss of pigmented neurons in the SN with gliosis. Other areas in the brainstem were reported to be unaffected. Importantly, no deposition of α-synuclein or Tau was observed in the brain lesions. Another neuropathological report of a patient with the R1441G mutation also found moderate loss of pigmented neurons in the SNpc [68]. Other brainstem nuclei were not affected and no α-synuclein aggregates were observed, which was consistent in these two neuropathological reports.

### 3.7. R1441H Mutation

The R1441H mutation was first found in North America in 2005 by Zabetian and colleagues (Family B; [36]). The age at onset of the proband was 55 years. The proband has an unaffected sibling carrying the mutation who was younger than the maximum age of onset in this family. Mata and colleagues also found the mutation in a Taiwanese family (Family TA; [35]). The clinical details of Family TA were reported in the follow-up study [69]. In Family TA, there were three affected individuals carrying the heterozygous R1441H mutation. Their ages at onset were 58, 47, and 49 years, which is slightly younger than that of the typical PD. The clinical course was similar to that of typical PD, with a good early response to levodopa. The proband manifested several psychological complications, including dementia at 73 years of age, but the other two affected individuals did not at the time of examination in their 50s.

Familial cases with a follow-up report have also been found in Portugal [70,71]. The original report described a Portuguese family (Lisb-F2) with two affected individuals carrying the R1441H mutation with ages at onset of 32 and 57 years [70]. No cognitive impairment was observed. There have also been a number of reports describing the familial cases inheriting the R1441H mutations worldwide [3,26,31,60,72,73,74,75,76,77], supporting the pathogenicity of the R1441H mutation. Some of these families had different haplotypes [78], further confirming the high frequency of the mutation at the Arg1441 codon. Overall, the age at onset in PD patients with the R1441H mutation varied from the 40s to the 60s, but the clinical course was indistinguishable from that of sporadic PD. There have been a few, but much less frequent, reports describing the R1441H mutation in sporadic PD patients when compared to the R1441C/G mutations [46,79], suggesting a more robust penetrance of this mutation.

Neuropathological examination of autopsied brains of patients with the R1441H mutation was reported by Takanashi and colleagues [74]. They reported two Japanese consanguineous families with the R1441H mutation and eight affected individuals. Of the eight patients, five had the homozygous R1441H mutation and the rest had the heterozygous mutation. The mean age at onset for homozygotes and heterozygotes was 61.60 ± 7.23 and 68.50 ± 6.54 years, respectively, with no statistical difference, suggesting that the dose of the R1441H mutation has no effect on age at onset. Two homozygotes and one heterozygote underwent neuropathological examination. These patients consistently showed levodopa-responsive late-onset parkinsonism without cognitive decline or autonomic dysfunction. Neuropathological examination revealed severe loss of dopaminergic neurons and astrogliosis in the SNpc but not in the LC. Lewy pathology was not found in any part of the brains. All three patients had primary age-related tauopathy and amyloid plaque deposition within the normal aging range. These results suggest that the R1441H mutation causes isolated nigral degeneration without α-synuclein or Tau deposition.

### 3.8. R1441S Mutation

An FPD family inheriting the R1441S mutation was reported by Mata and colleagues in 2016 [80]. There were six affected individuals in the family spanning three generations, three of whom were genetically analyzed and found to be heterozygous for the R1441S mutation. The ages at onset of the three individuals were 45, 68, and 76 years. Their symptoms began with asymmetric resting tremor that responded well to anti-parkinsonian treatment. They also had mild cognitive impairment at or just before the onset of motor symptoms.

Although only one familial case has been reported, the R1441S mutation appears to be pathogenic because three of the mutations described above (i.e., R1441C, R1441G, and R1441H) have been found in the same residue and the R1441S mutation cosegregates well with the disease.

### 3.9. Y1699C Mutation

The large German-Canadian FPD family (Family A) with autosomal dominant inheritance was first described in 1997 by Wszolek and colleagues [81]. In the paper identifying *LRRK2* as the causative gene for PARK8, they identified the Y1699C mutation in Family A and provided statistical evidence for the pathogenicity of this mutation [13]. Family A contained 12 affected members with a mean age at onset of 53 years (35–65 years), slightly younger than that of sporadic PD. They reported levodopa-responsive parkinsonism, and two patients showed dementia. They performed biochemical and neuropathological examination on two non-demented individuals. In both cases, there was a marked reduction in striatal dopamine levels. Histochemistry showed severe loss of pigmented neurons in the SN, but there were no LBs. One of the patients showed amyloid deposition consistent with mild to moderate Alzheimer’s disease, while the other showed mild neurodegeneration in the anterior horn of the spinal cord consistent with motor neuron disease. They observed abundant eosinophilic granules in the surviving neurons in the SN and LC, although the details of the granules were not clarified.

There is also a large British family, the Lincolnshire kindred, who inherit the Y1699C mutation first described by Nicholl and colleagues in 2002 [82]. To date, 25 affected members of the Lincolnshire kindred have been reported, with a clinical course similar to typical PD, including the levodopa-responsive parkinsonism and no cognitive impairment. The authors noted that the mean age of onset in the living subjects was 57 years (44–72 years), which is typical for sporadic PD. It has been noted that the mean age at onset tends to decrease with each generation [83]. The Y1699C mutation was found in another paper first identified *LRRK2* as the gene responsible for PARK8 [12].

Later, the brain pathology of a member of the Lincolnshire kindred was reported [83]. The clinical course of this patient was typical of PD with orthostatic hypotension, although the age of onset was 50 years. Histopathological examination revealed severe loss of pigmented neurons with marked gliosis in the SN and LC. A small amount of Lewy pathology was observed in the SN, LC, and olfactory bulb. There were also a few LBs in the neocortex. Neurofibrillary tangles were seen in the hippocampus and entorhinal cortex, but not in the brainstem. Occasional Aβ deposits were present in the neocortex. These observations suggest that the Y1699C mutation causes a severe loss of dopaminergic neurons in the brainstem with diverse neuropathology.

### 3.10. G2019S Mutation

Shortly after the first two reports of the mutations in the *LRRK2* gene responsible for FPD, one of the groups identified the G2019S mutation in two American FPD families [84]. One of the families was of English ancestry (Family 292), whereas the other family was Ashkenazi Jewish (Family 415). The mean age at onset of the five affected individuals in Family 292 was 58.4 years (45–73 years), whereas only one affected individual was studied in Family 415, whose age at onset was 54 years. The clinical phenotype of these patients was similar to that of typical PD. Haplotype analysis suggested that these two families shared a common founder.

Subsequently, the G2019S mutation was found in PD patients independently by three groups [85,86,87]. Nichols and colleagues screened for the G2019S mutation in Caucasian FPD cases and found this mutation in 5% of familial cases, with one homozygous case [85]. The mean age at onset of the G2019S carriers was 61.1 ± 13.9 years. Although patients with the G2019S mutation exhibited typical PD symptoms, the authors noted that carriers of the mutation tended to have a longer disease duration but milder symptoms, suggesting that the mutation was associated with slower disease progression. A similar difference between G2019S-associated and idiopathic PD was also noted in a paper by Healy and colleagues [88]. Gilks and colleagues found the G2019S mutation in eight sporadic PD patients [86]. Histopathological examination revealed neuronal loss and Lewy pathology in the SN in three cases. The mean age at onset was 57.4 years (41–70 years), and the clinical symptoms were similar to those of typical PD, including levodopa-responsive parkinsonism and treatment-related dyskinesia. In the article by Di Fonzo and colleagues, they identified two Italian, one Portuguese, and one Brazilian family inheriting G2019S, suggesting that this mutation occurs worldwide [87]. The mean age at onset was 50.5 years (38–68 years). Apart from the wide range of ages at onset, all patients carrying the G2019S mutation presented with symptoms that were clinically indistinguishable from sporadic PD.

Familial cases carrying the G2019S mutation were also reported by several groups in 2005 [24,35,36,83,89,90,91,92]. Paisán-Ruíz and colleagues reported an Anglo-Saxon family (Family PD2) in which they identified nine patients carrying the mutation [24]. Although their clinical manifestations appeared similar to those of typical PD, they also noted the variation in age at onset (41–80 years). Kachergus and colleagues reported 13 families of North American and European descent [89], while Lesage and colleagues reported five families of European descent and seven families of North African descent [91]. Surprisingly, the families in these two reports shared the common disease-linked haplotype, suggesting that the mutation occurred in a common ancestor living in the 13th century [91].

Among the LRRK2 mutations, the G2019S mutation was found to be the most common. Ozelius and colleagues screened 120 Ashkenazi Jewish patients with PD and found the G2019S mutation in 22 patients (18.3 percent) [93]. Lesage and colleagues also reported a high prevalence in North African Arabs, where they screened 59 patients with PD and detected the G2019S mutation in 23 patients (39 percent) [94]. A multicenter analysis of the prevalence of the G2019S mutation in PD patients showed that the prevalence is highest in North African Arabs and Ashkenazi Jews, whereas the G2019S mutation is rare in Asians [88] (reviewed in [95]).

In addition to the variable age at onset in patients carrying the G2019S mutation, the G2019S mutation has been shown to have reduced penetrance, meaning that there are more mutation carriers, when compared to other LRRK2 mutations, who do not develop PD throughout their lives. In a 2011 paper by Goldwurm, the penetrance of the G2019S mutation was estimated to be 33% at 80 years of age [96]. Marder and colleagues reported a similarly low penetrance of the G2019S mutation in Ashkenazi Jews, which was 26% at 80 years of age [97]. Lee and colleagues also reported that the penetrance of the G2019S mutation was 42.5% at 80 years in Ashkenazi Jews and 25% at 80 years in non-Ashkenazi Jews [98]. These results suggest that the pathogenicity of the G2019S mutation is more likely to be influenced by other factors, including genetic factors other than the LRRK2 mutation, as well as environmental factors [66].

Many neuropathological studies of PD patients with the G2019S mutation have been reported since the initial report by Gilks and colleagues as described above [86]. Ross and colleagues reported eight autopsied cases carrying the G2019S mutation [99]. The mean age at onset was 63 years (41–79 years). All cases showed Lewy pathology in the brainstem and in some cases in the neocortex. Giasson and colleagues reported three autopsied cases with the G2019S mutation [100]. The ages at onset were 47, 76, and 59 years, and the clinical course was similar to that of typical PD. Severe loss of pigmented neurons in the SNpc and LC was observed in all cases, and two of them showed typical Lewy pathology confined to the brainstem nuclei. Notably, one case showed no Lewy pathology anywhere in the brain. The absence of Lewy pathology was also reported by Rajput and colleagues, who found prominent Tau pathology consistent with progressive supranuclear palsy (PSP) [101]. Gaig and colleagues reported two autopsied cases with the G2019S mutation [102,103]. The ages of onset were 61 and 63 years. In one case they found severe neuronal loss in the SN and LC with extensive Lewy pathology, whereas in the other case there was mild neuronal loss in the SN and LC without Lewy pathology. Poulopoulos and colleagues have also documented the similar variation in Lewy pathology in G2019S-associated PD patients [104]. Henderson and colleagues presented the results of a relatively large set of neuropathological findings in PD patients with the G2019S mutation [105]. They found that 55.5% of the cases with G2019S showed Lewy pathology in various regions of the brain, including the midbrain; more importantly, they found that 100% of the cases were positive for Tau pathology in the limbic areas, including the amygdala and hippocampus. Deposition of TAR DNA-binding protein 43 (TDP-43) in the brains of individuals carrying the G2019S mutation has also been documented in a few papers [106,107,108]. Their clinical manifestations were frontotemporal lobar degeneration (FTLD), PD, and dysphagia. Collectively, the results of these neuropathological studies suggest that the G2019S mutation causes typical PD pathology with neuronal loss in the SN and LC accompanied by Lewy pathology in most cases, although the G2019S mutation may cause deposition of other amyloidogenic proteins such as Tau and TDP-43, leading to neurodegeneration independent of α-synuclein and clinical manifestations different from PD.

### 3.11. I2020T Mutation

A large family with inherited PD was found in Japan (Sagamihara kindred; [109]). Neuropathological examination of affected members revealed mild to moderate loss of pigmented neurons in the SN, but the LC was preserved. Lewy pathology was not present in any part of the brain. The chromosomal locus linked with the Sagamihara kindred was identified by linkage analysis on chromosome 12p11.2-q13.1 and named PARK8 [110].

The I2020T mutation was subsequently identified in a German family (Family 32; [13]). Family 32 had three affected individuals with the I2020T mutation, whose clinical course resembled that of typical PD, with mean age at onset of 54 years (48–59 years). There has been no follow-up report for Family 32 to date. The same mutation was subsequently identified in the Sagamihara kindred [111]. The mean age at onset is 56 years (38–74 years), as detailed in the most recent report [112]. Their clinical features were indistinguishable from typical idiopathic PD, but they noted that their autonomic symptoms were milder than sporadic PD and that most affected individuals remained cognitively intact.

Other FPD families with the I2020T mutation have been found in the Japanese population [75,113], but the haplotype analysis revealed that these families share a common founder with the Sagamihara kindred, suggesting that they are distant relatives. The occurrence of the I2020T mutation appears to be extremely rare, as only two founders have been identified to date, and there are few findings in case-control studies. However, the independent occurrence of the I2020T mutation in the German Family 32 and the Sagamihara kindred was confirmed by haplotype analysis [114]. For the Sagamihara kindred, it has been suggested that the mutation occurred approximately 1300 years ago [75].

Results of neuropathological analysis have been reported from the Sagamihara kindred [112,115]. In the earlier report by Hasegawa and colleagues, they reported the results of eight subjects. Six of the eight subjects showed mild neuronal loss in the SNpc but not in the LC. Of note, the neuronal loss was more severe in the substantia nigra pars reticulata, which is not usually seen in sporadic PD. Also atypical was the absence of Lewy pathology throughout the brain. In one case, there were α-synuclein-positive glial cell inclusions (GCIs) in the putamen, and the case was neuropathologically diagnosed as multiple system atrophy with parkinsonism (MSA-P). Only one of the eight cases examined showed extensive α-synuclein-positive Lewy pathology in the SN, LC, dorsal motor nucleus of the vagal nerve, and raphe nuclei. Neuronal loss was also mild in this case. The latter report by Ujiie and colleagues added an additional case [115] whose autopsy also showed marked neuronal loss in the SN with well-preserved neurons in the LC and no Lewy pathology, consistent with the earlier report. Interestingly, they also noted the deposition of hyperphosphorylated Tau in some cases. In conclusion, the extent of the neuropathological pleomorphism with the I2020T mutation seems to be similar or even more pronounced than with the G2019S mutation.

## 4. Effects of the Pathogenic Mutations on the Functions and Molecular Properties of LRRK2

LRRK2 contains a Ras-of-complex proteins (ROC) domain and a Ser/Thr protein kinase domain within a single polypeptide (Figure 1A). The ROC domain has been shown to bind guanine nucleotides, which is critical for the kinase activity of the same molecule [116,117]. The ROC domain is followed by the carboxy-terminal of ROC (COR) domain, the function of which is unknown. The ROC-COR tandem is a domain architecture shared by the ROCO protein family [118]. Therefore, the COR domain is thought to play an important role in the function of the ROC domain. The pathogenic mutations described above are located in the ROC-COR-kinase domains (Figure 1A), suggesting that the pathogenic mutations cause PD by affecting the functions and/or molecular properties of these domains.

### 4.1. In Vitro Kinase Activity and Cellular Substrate Phosphorylation Activity

LRRK2 has been reported to phosphorylate a number of proteins, including myelin basic protein (MBP) [119], Ezrin/Moesin/Radixin (ERM) family proteins [120], 4E-BP1 [121], the ribosomal protein s11/s15/s27 [122], p62/SQSTM1 [123], and small GTPase Rab proteins [14]. LRRK2 also phosphorylates itself (i.e., autophosphorylation) on a number of residues, mainly around the ROC domain [124,125,126,127,128]. Peptide substrates for LRRK2 (e.g., LRRKtide and Nictide) have also been developed for the quantification of in vitro kinase activity [120,129]. West and colleagues first reported the establishment of the in vitro LRRK2 kinase assay in 2005 [119]. In this paper, they described that the G2019S mutation, but not the R1441C mutation, increased the phosphorylation of MBP 3-fold, and that the autophosphorylation of LRRK2 was increased 2.5-fold and 1.5-fold by the G2019S and R1441C mutations, respectively. The increase in in vitro substrate phosphorylation by the G2019S mutation has been confirmed by many researchers using different assay systems and substrates, and there is now consensus that the G2019S mutation upregulates the in vitro kinase activity of LRRK2 2–3-fold.

The effect of other mutations on in vitro kinase activity appears to be small and is variable between reports. For example, Gloeckner and colleagues reported a 1.5-fold increase in LRRK2 autophosphorylation by the I2020T mutation [130], whereas Jaleel and colleagues reported a 50% decrease in LRRK2 autophosphorylation by the same mutation [120]. We also observed a significant decrease in LRRK2 autophosphorylation by the I2020T mutation, although the order of magnitude was very small [125]. Such variation between studies is largely due to the differences in assay systems, including protein sources, detection systems, substrates used, etc. Now that Rab proteins have been identified as physiologically relevant substrates (see below), we should try to standardize the method of the in vitro LRRK2 kinase assay using Rab proteins as authentic substrates. Autophosphorylation at Ser1292 is considered to be the most physiologically relevant and the only site observed in cultured cells [126]; all mutations, including I2020T but not Y1699C, increased the autophosphorylation at Ser1292 in cultured cells [126].

Steger and colleagues were the first to identify the physiological substrates of LRRK2, the Rab proteins [14]. In the paper published in 2016, they showed that the G2019S mutation increased Rab8A phosphorylation 2-fold, while the R1441C mutation did not change it in an in vitro assay [14]. However, by overexpressing both LRRK2 and substrate Rab proteins in cultured cells, they observed a significant upregulation of Rab phosphorylation by all pathogenic mutations, which was further confirmed in mice with knock-in R1441G or G2019S mutations [16]. The magnitude of the increase was even more remarkable for mutations other than G2019S, suggesting that the mechanism of upregulation of substrate phosphorylation by non-G2019S pathogenic mutations is independent of their effect on in vitro kinase activity. Indeed, in humans, Rab10 phosphorylation was increased in peripheral blood neutrophils obtained from R1441G carriers but not from G2019S carriers [131]. Recently, Kalogeropulou and colleagues reported the results of a large-scale analysis investigating the effects of LRRK2 variants on its substrate phosphorylation [132]. Their results also showed that all pathogenic mutations increased the cellular phosphorylation of Rab10 Thr73. It remains unclear why Rab8A phosphorylation was not increased by the non-G2019S mutations in the in vitro kinase assay, whereas it was dramatically increased in vivo. Given that Rab proteins are lipid-modified and anchored to membranes in cells but not in in vitro assays, it may be interesting to investigate whether the subcellular location (i.e., membrane vs. cytosol) is involved in the increase in substrate Rab phosphorylation by the non-G2019S mutations.

In summary, the pathogenic mutations of LRRK2 upregulate its substrate phosphorylation in cells and tissues (Table 2), but the mechanisms underlying the upregulation appear to be different between G2019S and non-G2019S mutations.

### 4.2. In Vitro GTPase Activity

The ROC domain of LRRK2 binds GTP and GDP and has in vitro GTP hydrolyzing (GTPase) activity. Since some of the pathogenic mutations of LRRK2 occur in its ROC domain, the effect of these mutations on the GTPase activity was investigated. In our paper published in 2007, we did not detect GTPase activity in an in vitro assay using full-length LRRK2 immunoprecipitated from cells overexpressing LRRK2 [116]. However, Lewis and colleagues have shown that LRRK2 immunoprecipitated from cultured cells has an in vitro GTPase activity that is attenuated by the R1441C mutation [133]. The discrepancy in the ability to hydrolyze GTP may also be due to the different experimental settings: we examined the amount of GTP/GDP bound to LRRK2 after the GTPase reaction, whereas other groups examined the amount of GTP/GDP in the whole reaction mixture, for example.

The effect of pathogenic LRRK2 mutations on its GTP-binding activity appears to be variable. The first report by West and colleagues in 2007 showed that the I1371V, R1441C/G, and Y1699C mutations upregulate GTP-binding activity [117], whereas we observed a similar increase in GTP binding by the R1441C mutation, but not by the Y1699C mutation [125]. In the paper by Lewis and colleagues, there was no change in GTP-binding activity due to the R1441C mutation [133]. These early observations suggest that the difference in GTP-binding activity caused by the pathogenic mutations may exist but is relatively small and can vary depending on the experimental settings. According to the recently solved structures of LRRK2, these pathogenic mutations in the ROC-COR domains are located at the interface of the ROC and COR domains and affect their interaction. Further discussion can be found in Section 4.3.

The effect of pathogenic mutations on GTPase activity has also been extensively examined and, as mentioned above, an early observation was that in vitro GTPase activity of immunoprecipitated full-length LRRK2 was decreased by the R1441C mutation. Guo and colleagues also showed that the R1441C mutation attenuated in vitro GTPase activity using immunoprecipitated full-length LRRK2 [134]. Li and colleagues showed in vitro GTPase activity using full-length LRRK2 immunoprecipitated from BAC transgenic mouse brains as well as ROC recombinant proteins; in the latter experiments they reported a ~30% reduction in in vitro GTPase activity by the R1441C/G mutations [135]. Xiong and colleagues reported that the R1441C/G and Y1699C mutations slightly reduced the in vitro GTPase activity to 80–90% when compared to wild-type (WT) when using immunoprecipitated full-length LRRK2 [136].

A more recent report by Liao and colleagues in 2014 has systematically characterized the biochemical and biophysical properties of the recombinant ROC protein and elucidated that the R1441H mutation decreased in vitro GTPase activity, but did not alter its oligomeric state, binding affinity to GTP/GDP, or overall conformation [137]. The same group also investigated the effects of the R1441C/G and N1437H mutations using similar experimental batteries and showed a dramatic reduction in in vitro GTPase activity by these mutations [138,139].

Based on these observations, it has been hypothesized that the pathogenic LRRK2 mutations in the ROC and COR domains attenuate its GTPase activity to increase GTP-bound LRRK2, thereby upregulating its kinase activity. However, the upregulation of LRRK2 kinase activity in vitro by these mutations has not been consistently demonstrated, as mentioned in the previous section. Thus, a classic question still remains: how do changes in the ROC domain affect substrate phosphorylation? If changing the structural linkage between the ROC and kinase domains activates the latter, why is that activation has not recapitulated in in vitro kinase assays? Alternatively, if the ROC mutations alter the biological properties of LRRK2 (e.g., subcellular localization) and increase the probability that substrate Rab proteins are in the vicinity of LRRK2, how do they alter its biological property? Further investigation addressing the structural basis of the regulation of kinase activity by the ROC domain and the subcellular localization of mutant LRRK2 is required to clarify these points.

### 4.3. Three-Dimensional Structure

Since GTP binding to the ROC domain of LRRK2 is critical for the kinase activity of the same molecule, it is reasonable to assume that the ROC domain interacts with the kinase domain to regulate its function. The three-dimensional structure of full-length LRRK2 was recently solved by cryogenic electron microscopy (cryo-EM) (Figure 2A–C) [21]. Although the structure of the kinase domain was apparently in its inactive form, the overall domain architecture clearly showed that there are interdomain interactions between the ROC and COR domains, and between the ankyrin repeat (ANK)/leucine-rich repeat (LRR) domain and the kinase (KIN) domain. The residues mutated in FPD, namely Asn1437, Arg1441, and Tyr1699, were all located in the former interface (Figure 2D), suggesting the importance of this interaction in the phosphorylation of substrate proteins. More recently, a preprint published by the same group in bioRxiv showed the active conformation of full-length LRRK2 [140]. They proposed that the substitution of Asn1437 and Arg1441 causes the ROC and COR domains to favor the active conformation. In this model, Tyr1699 is located at the junction of the ROC and COR domains. Therefore, substitution with less bulky Cys might facilitate the transition from the inactive to the active conformation. Furthermore, Ile2020 was located in the hydrophobic and hydrophilic environments in the inactive and active states, respectively. I2020T replaces a hydrophobic residue with a hydrophilic residue. Therefore, the I2020T mutation will stabilize the active conformation. For the G2019S mutation, a more localized conformational change within the kinase domain may lead to an increase in intrinsic kinase activity.

Collectively, the non-G2019S pathogenic mutations could lead to increased substrate phosphorylation by altering the overall domain structure, which plays a role in switching between the inactive and active states. Further studies are needed to determine whether the reduction in GTPase activity due to the mutations described in the previous section has an additive effect on the change in domain structure.

### 4.4. Phosphorylation of LRRK2 and Interaction with 14-3-3

It has been reproducibly observed that LRRK2 is phosphorylated at Ser910 and Ser935 under physiological conditions [117,141], for which the responsible kinase(s) remains to be elucidated. Interestingly, these sites are dephosphorylated upon treatment with LRRK2 inhibitors by unknown mechanisms [142,143] and have been used as surrogates for target engagement of LRRK2 inhibitors in animals and humans [144,145]. The pathogenic LRRK2 mutations differentially regulate Ser910/935 phosphorylation. Nichols and colleagues showed that Ser910/935 phosphorylation was downregulated by the R1441C/G/H, Y1699C, and I2020T mutations, but not by the G2019S mutation [141]. Similar observations were later made by other researchers [146,147]. Doggett and colleagues found that LRRK2 is also phosphorylated at Ser955 and Ser973, and these two additional phosphosites are regulated by pathogenic LRRK2 mutations in a manner similar to Ser910/935 [147].

Ser910/935 phosphorylation is required for the interaction of LRRK2 with 14-3-3 family proteins [141]. The biological significance of the interaction between LRRK2 and 14-3-3 is still unclear, but it has been suggested that loss of binding to 14-3-3 alters the subcellular localization of LRRK2 from a diffuse pattern to punctate/aggregate or microtubule-like/filamentous structures in the cytoplasm [141,147]. Kett and colleagues have shown that R1441C/G, Y1699C, and I2020T, but not G2019S, mutations in LRRK2 promote its formation of filamentous structures [148], which is consistent with the observation that these mutations are less phosphorylated at Ser910/935 compared to WT and G2019S LRRK2.

The three-dimensional structure of LRRK2 filaments has recently been solved by cryo-electron tomography (cryo-ET) [149] and cryo-EM [150]. Full-length LRRK2 forms double-stranded right-handed helices around microtubules, presumably through the homotypic interaction between WD40 domains [149]. Deniston and colleagues showed that a carboxy-terminal half of LRRK2 consisting of the ROC-COR-KIN-WD40 domains binds to microtubules and inhibits the motility of kinesin and dynein [150]. Further studies are needed to elucidate how pathogenic LRRK2 mutations affect its inhibitory role on motor proteins.

### 4.5. Interaction with Other Binding Partners

In addition to the 14-3-3 proteins, several other proteins have been shown to interact with LRRK2 in cultured cells. LRRK2 binding proteins, for which the direct interaction has been shown to be altered by pathogenic mutations of LRRK2, were searched in the IntAct database (https://www.ebi.ac.uk/intact/; last access date: 14 May 2023). The database search retrieved an Fas-associated protein with death domain (FADD) [151], dishevelled family proteins (DVL1–3) [152], MAP kinase kinases (MKK6/7) [153], Rac1 [154], Akt1 [155], β-tubulin [156], protein phosphatase 1α (PP1α) [157], and protein kinase A regulatory subunit IIβ (PKARIIβ) [158]. The effects of pathogenic LRRK2 mutations on these interactions are quite variable and their pathological relevance remains unclear (Table 3). To elucidate the significance of the changes in the interactions, further studies addressing how these interactions affect LRRK2-mediated Rab phosphorylation are also needed.

Another important binding partner of LRRK2 is the Rab32 subfamily, namely Rab29, Rab32, and Rab38 [159], as Rab29 is a known activator of LRRK2 [160,161]. Purlyte and colleagues found that substitutions in the ANK of LRRK2 inhibit activation by Rab29 in cells, suggesting that Rab29 binds to the ANK [160], while McGrath and colleagues used purified proteins to show that Rab29/32/38 bind directly to the ARM (1-552 a.a.) of LRRK2 [159]. Although the binding site of Rab29 in LRRK2 remains controversial, pathogenic LRRK2 mutations may indirectly enhance the binding of Rab29 to the amino-terminal part of LRRK2 and accelerate the phosphorylation of the substrate Rab proteins. Importantly, Purlyte and colleagues showed that Rab29 was still able to activate pathogenic LRRK2 mutants [160], suggesting that the mutations do not convert LRRK2 to a constitutively activated form.

## 5. Implications for PD Therapy, Diagnosis, and Patient Management

The clinical, pathological, and molecular discoveries related to pathogenic LRRK2 mutations have provided much information to establish therapy and diagnostics for both LRRK2-associated and sporadic PD.

The uniform increase in substrate phosphorylation by pathogenic LRRK2 mutations suggests that a therapy that reduces substrate phosphorylation should be beneficial for PD patients with LRRK2 mutations. In addition, Rab10 phosphorylation has been shown to be increased in the brains of sporadic PD patients [162], suggesting that reducing substrate phosphorylation of LRRK2 may also be beneficial for sporadic PD patients. Indeed, small molecule LRRK2 inhibitors and antisense oligonucleotides that reduce the expression of LRRK2 are in clinical trials as potential drugs for PD, some of which have completed phase 1/1b and are moving into later stages [145,163]. Small molecules that specifically inhibit G2019S LRRK2 have also been reported [164,165]. Considering that the G2019S mutation occurs heterogeneously in most cases, the use of G2019S-specific inhibitors in PD patients with the G2019S mutation may be better than non-specific LRRK2 inhibitors to avoid possible adverse effects caused by excessive inhibition of LRRK2 kinase activity. There will also be specific inhibitors for non-G2019S mutations once the structural basis for the increase in substrate phosphorylation by non-G2019S mutations is elucidated.

As described in Section 3, the penetrance of the pathogenic LRRK2 mutations is sometimes low, and genetic tests that detect the mutations are not always useful in predicting who will develop PD. Because disease-modifying therapies based on LRRK2 inhibition are not yet available to date, genetic testing should be used only for genetic counseling of patients and their families in familial cases and for academic studies [166]. Since individuals with pathogenic LRRK2 mutations are more likely to develop PD than the general population, it may be important to establish cohorts based on genetic testing for prospective clinical/biomarker studies of asymptomatic LRRK2 mutation carriers and clinical trials for secondary prevention once a potential disease-modifying therapy becomes available. Similar efforts have been made in drug development for Alzheimer’s disease (e.g., DIAN (The Dominantly Inherited Alzheimer Network)) [167,168]. Clearly, further research to establish body fluid biomarkers and diagnostic imaging for PD is a prerequisite for such a study, which will also be facilitated by studies in LRRK2-associated PD patients.

## 6. Conclusions

The fact that the pathogenic mutations of LRRK2 uniformly increase substrate phosphorylation of LRRK2 strongly suggests that it is the increase in substrate phosphorylation that leads to neurodegeneration in LRRK2-associated PD. It is highly likely that Rab proteins are the key substrates for this process, but the mechanism by which Rab phosphorylation causes neurodegeneration remains elusive. The G2019S mutation has a milder effect than other mutations in cultured cells, knock-in mice, and human carriers. This may be related to the fact that the clinical symptoms of G2019S carriers tend to be milder than typical PD and that the penetrance of the disease in G2019S carriers is lower than in carriers of other pathogenic mutations.

On the other hand, the cause of the diversity in pathology is unlikely to be related to increased substrate phosphorylation. It is reasonable to assume that factors other than the LRRK2 mutation are strongly involved in the development of each pathology, since there is no one-to-one correlation between mutation and pathology. The neuropathological diversity seems to be more pronounced for the kinase domain mutations (i.e., G2019S and I2020T) (Table 1). There may be genetic factors that modify the pathology produced by the G2019S and I2020T mutations, such as DNM3 and VAMP4, which have been suggested to modify the age of onset and risk of developing PD by the G2019S mutation. It is also possible that, in neurodegeneration caused by hyperphosphorylation of LRRK2 substrates, it is a stochastic process as to which pathology develops. To identify the molecular entities responsible for the differences in pathology, studies using multi-omics analysis of, for example, patient samples from the same family with different pathologies would be required.

Elucidating the mechanisms underlying the formation of pathological diversity is expected to contribute greatly to the elucidation of the pathophysiology of not only LRRK2-associated PD but also other neurodegenerative diseases.

## Figures and Tables

**Figure 1 biomolecules-13-00845-f001:**
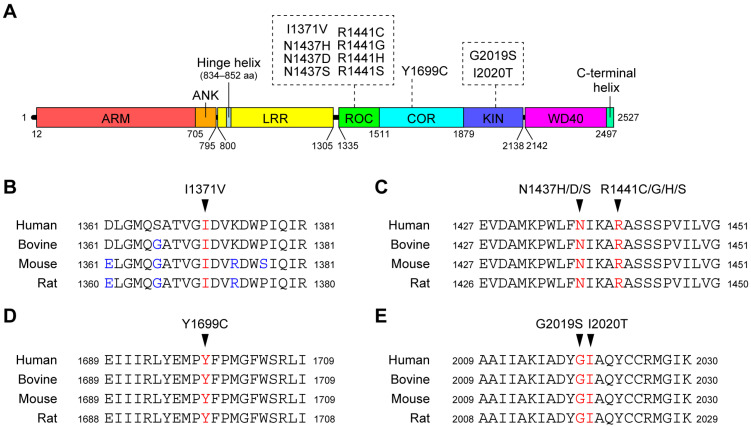
(**A**) Schematic representation of the domain architecture of LRRK2 and pathogenic mutations. Numbers represent amino acid numbers from the amino terminus. The domain boundaries correspond to a paper reporting the three-dimensional structure of full-length LRRK2 using cryo-electron microscopy [21]. (**B**) Conservation of amino acid sequences around the I1371V (**B**), N1437H/D/S and R1441C/G/H/S (**C**), Y1699C (**D**), and G2019S and I2020T (**E**) mutations between species. The mutated residues are shown in red, and amino acids that differ from human LRRK2 are shown in blue.

**Figure 2 biomolecules-13-00845-f002:**
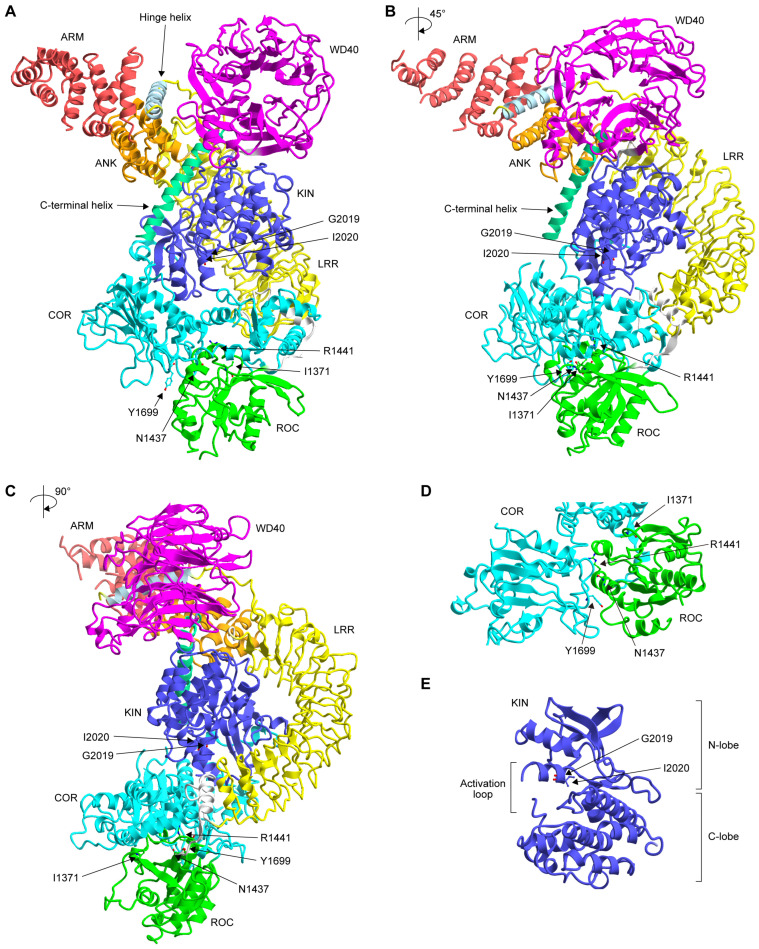
(**A**–**C**) Three-dimensional structure of full-length LRRK2 (PDB: 7LHW) solved by cryo-EM at a resolution of 3.1 Å. The color of each domain was made according to the scheme shown in Figure 1A. Note that the structure of the amino-terminal 557 residues was flexible and not solved in this structure. The 3D structures in (**B**,**C**) are 45° and 90° rotations of the 3D structure in (**A**), respectively. (**D**,**E**) Close-up of the ROC-COR domain interface (**D**) and kinase domain (**E**). The mutated residues are indicated by a ball-and-stick representation. The structural representations were displayed using CueMol2 (http://www.cuemol.org; last access date: 14 May 2023).

**Table 1 biomolecules-13-00845-t001:** Clinical and pathological features of patients with LRRK2 pathogenic mutations.

Mutations	Clinical Symptoms	Pathology
I1371V	Typical PD but younger onset.	1 case reported. Neuronal loss in the SN and LC accompanied by LBs in the SN, LC, and cerebral cortex.
N1437H	Typical PD but younger onset.	1 case reported. Neuronal loss in the SN and LC accompanied by LBs in the SN and LC.
N1437D	Younger onset.	N/A
N1437S	N/A	N/A
R1441C	Typical PD.	4 cases reported. Neuronal loss in the SN and LC in all cases. LBs in 2 cases, no LBs with Tau inclusions in 1 case, no LBs or Tau inclusion in 1 case.
R1441G	Typical PD with less autonomic and cognitive impairment. Onset age is typical in most cases, but younger in one Japanese family.	2 cases reported. Moderate neuronal loss in the SN but not in the LC. No LBs in both cases.
R1441H	Typical PD. Variation in age at onset.	3 cases reported (2 homozygotes and 1 heterozygote). Neuronal loss in the SN but not in the LC. No LBs in any cases.
R1441S	Typical PD, but cognitive impairment was noted before motor symptoms.	N/A
Y1699C	Typical PD but younger onset.	3 cases reported. Neuronal loss in the SN without LBs (2 cases). Neuronal loss in the SN and LC with mild LBs (1 case).
G2019S	Milder than typical PD with slow progression. Low disease penetrance was noted. Variable age at onset.	Many cases reported. Neuronal loss in the SN in all cases, but the existence of LBs was extremely variable. Also, Tau inclusions without LBs in many cases. TDP-43 accumulation was also noted in some cases.
I2020T	Typical PD but younger onset.	9 cases reported (all from the same family). Neuronal loss in the SN, but the LC was largely spared in all cases. No LBs in 7 cases. GCIs in 1 case consistent with MSA. Typical LBs in 1 case. Tau inclusions in the affected areas in 4 cases.

N/A: Not available.

**Table 2 biomolecules-13-00845-t002:** Effects of pathogenic mutations on functions of LRRK2.

Mutations	In Vitro Kinase Activity	Cellular Substrate Phosphorylation	In Vitro GTPase Activity
I1371V	No consistent changes	N/A	N/A
N1437H	No consistent changes	Increase	Decrease
N1437D	No consistent changes	N/A	N/A
N1437S	No consistent changes	N/A	N/A
R1441C	No consistent changes	Increase	Decrease
R1441G	No consistent changes	Increase	Decrease
R1441H	No consistent changes	Increase	Decrease
R1441S	No consistent changes	Increase	N/A
Y1699C	No consistent changes	Increase	Decrease
G2019S	Increase	Increase	N/A
I2020T	No consistent changes	Increase	N/A

N/A: Not available.

**Table 3 biomolecules-13-00845-t003:** Effects of pathogenic mutations on interaction with binding partners.

Binding Partners	R1441C/G/H	Y1699C	G2019S	I2020T	References
FADD	↑	↑	↑	↑	[151]
DVL1	↑	↓	N.D.	N.D.	[152]
DVL2/3	→	↓	N.D.	N.D.	[152]
MKK6	↑	→	→	→	[153]
MKK7	↑	N.D.	→	N.D.	[153]
Rac1	↓	↓	↓	↓	[154]
Akt1	↓	N.D.	↓	↓	[155]
β-tubulin	↑ (R1441C) ↓ (R1441G/H)	N.D.	N.D.	N.D.	[156]
PP1α	↑	↑	N.D.	↑	[157]
PKARIIβ	↓	N.D.	→	N.D.	[158]

↑: Increase; ↓: Decrease; →: No change; N.D.: Not determined.

## Data Availability

Data sharing not applicable.

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
