# Peer review of "Overview of the Impact of Pathogenic LRRK2 Mutations in Parkinson’s Disease"

_biomolecules, 2023, doi:10.3390/biom13050845_

Round 1

Reviewer 1 Report

The authors have outlined the different mutations in LRRK2 associated with PD, listing the occurrences and neuropathology from published studies.

Minor edits:

1)    Line 29: replace “akinesia” with “bradykinesia” since this is the appropriate symptom

2)    Line 29-30: The authors suggest that the listed symptoms occur as a result of neurodegeneration. However, this is not proven yet. These symptoms could occur as a result of intracellular pathological changes, without actual degeneration. It would be good to alter the sentence accordingly.

3)    Fig 1: Kindly re-check the boundaries for the protein domains on LRRK2, and re-label accordingly. The boundaries for ARM, ANK & LRR are off, as per published results.

4)    Fig 1: It might be helpful for readers to have the residues of the protein excerpts mentioned in the figures for each species. For example, in 1B, where does the DLGM… start on the human LRRK2 protein versus bovine, rat and mouse proteins. And where does ….PIQIR end for each of the species on their specific protein.

5)    Table 1: Please clarify what does “typical PD” or “typical motor symptoms of PD” refer to.

6)    Line 110-111: Please include reference for the inheritance pattern.

7)    In vitro GTPase activity: This section requires a bit more elaboration. The impact of GTPase activity on kinase activity of LRRK2 is unclear mainly because of the use of different substrates for assessing these activities between studies, and the different cellular platforms used to conduct these studies, etc. It would be helpful to include some of the authors’ insights in these sections.

8)    While the authors have done a great job at describing the clinical and neuropathological aspects of the different LRRK2 variants, the molecular sections lack the same detailing. It might be helpful if the authors could elaborate more in these sections, or at least refer to another recent review for more information on these sections.

Reviewer 2 Report

Title: Overview of the impact of pathogenic mutations in LRRK2.

In this review article, Genta Ito et al. summarized the pathogenic mutations in leucine-rich repeat kinase 2 (LRRK2) and their effects on patient's clinical symptoms and neuropathology. It mainly focuses on the pathological roles of LRRK2 mutations in Parkinson's disease (PD).

The authors provide a comprehensive review of important pathological mutations in LRRK2 and how these mutations affect the functions and molecular properties of LRRK2.

This manuscript is publishable in Biomolecules after major revisions.

Major comments are the following:

1.     This review focuses on the role of several LRRK2 mutations in PD. Are these LRRK2 mutations involved in other diseases? If so, the title of this manuscript is not appropriate. The authors should specify this point in the title.

2.     These mutations change the biochemical activities and structural features of LRRK2. Does LRRK2 have potential binding partners? If LRRK2 has other binding partners, for example, protein molecules, how do these pathological mutations affect LRRK2’s binding with other proteins?

3.     The structure of LRRK2 has been solved by cryo-EM (Myasnikov, A. et al. Cell 2021, 184, 3519-3527.e10). To help readers visualize the positions of these mutations and better understand how these mutations affect the structure of LRRK2, the authors should add one more figure, showing the structural model of LRRK2 and labeling these mutations in the protein structure, besides presenting the domain organization of LRRK2 in Figure 1.

Reviewer 3 Report

In this work, the authors have done a review of the impact of pathogenic mutations in LRRK2. They summarized the effects of pathogenic mutations in LRRK2 on clinical symptoms and the neuropathology of patients. They described each variant in detail. 

Critique

 The review is interesting and gives a nice view of the role of LRRK in PD, but to provide a comprehensive overview it lacks a more profound and deep discussion. It is written more in a form of a book chapter. 

Several studies have already been done on this topic. Why do the authors think their work is of unique value?

Turning this review article into a meta-analysis would be more useful and of higher scientific value.

In the discussion section, readers will benefit more, if the author can discuss more and summarize the potential role of variants in therapy, diagnostics, and management of patients with PD.

Round 2

Reviewer 2 Report

The overall quality of this manuscript is greatly improved after revision.